# OpenReview forum: "On the Resilience of LLM-Based Multi-Agent Collaboration with Faulty Agents"
_ICML.cc/2025/Conference — ICML 2025 poster_

### Official Review · Reviewer_3cYX · 2025-03-11

**Overall Recommendation:** 4

**Summary:**

The paper explores how well multi-agent systems based on large language models (LLMs) can handle errors introduced by faulty agents. It specifically looks at resilience across tasks like writing code, solving math problems, translating text, and evaluating written text. To test this, it uses two methods: AUTOTRANSFORM, which alters agent behaviors, and AUTOINJECT, which directly introduces errors into agent communications. Findings show that hierarchical system structures cope best with faults, outperforming flat and linear setups. Interestingly, linear structures were the least resilient. The authors recommend hierarchical models for designing robust multi-agent collaborations using LLMs.

**Claims And Evidence:**

Yes.

**Essential References Not Discussed:**

Need to supplement more discussions about published in 2024 and 2025, for example, the automated evaluation methods. Besides, in the introduction, I suggest the author could move the core claim at the very beginning. Since some of the basic claims are not so attractive recently but I like the basic idea.

**Ethical Review Concerns:**

No ethical concerns.

**Experimental Designs Or Analyses:**

I examined the experimental design and analyses conducted in this paper. The experiments utilized two well-defined error simulation methods, AUTOTRANSFORM and AUTOINJECT, across three clearly differentiated multi-agent system structures (linear, flat, and hierarchical).The metrics used (accuracy and specific task-related metrics such as Pass@1 and BLEURT) were appropriate and clearly presented. Overall, the experimental design and analyses are sound and rigorous.

The only concern here is the author only use several baseline like Self-Collab and CAMEL which cannot fully reflect the current advanced methods. Suggest to add more discussion about Graph-based framework (LangGraph, GPTSwarm), and most popular industrial practice ones like Dify, etc.

**Methods And Evaluation Criteria:**

Yes.

**Other Comments Or Suggestions:**

See my previous comments.

**Other Strengths And Weaknesses:**

- It would be beneficial if the authors could include additional experiments to demonstrate whether the observed issues persist across other frameworks.
- The idea of this paper is not brand-new but I appreciate the solid writing in this paper.

**Questions For Authors:**

See my previous comments.

**Relation To Broader Scientific Literature:**

The paper's key contributions connect closely with broader scientific literature on multi-agent systems, resilience, and safety in AI collaborations.

**Theoretical Claims:**

There are not many theoretical claims in this paper.

---

> ### Author Rebuttal · Authors · 2025-04-01
>
> We deeply appreciate reviewer 3cYX’s time for reviewing and your insightful comments. We are particularly encouraged that you find that the experiments and analyses are sound and rigorous. We address your concerns one by one:
>
> ## **[Design & Analysis] [W1]** Advanced methods like Graph-based frameworks.
>
> Inspired by GPTSwarm and MacNet, we design two advanced graph-based multi-agent frameworks using four agents:
>
> 1. **Complete graph (a flat structure)**: each agent generates their own answers. After receiving all others’ answers in the next run, they re-generate the answers after thinking. The final answer is the majority one.
>
> 2. **Star (a hierarchical structure)**: one leader proposes three approaches and distributes them to the three agents. After receiving the solutions, the leader gives its evaluation and generates the final answer.
>
> We evaluate GPT-3.5 using the Math task. The performance is shown in the table below. We can conclude that our methods and analyses are **applicable to diverse frameworks**. Flat structure still has **a lower performance** since there is no leader coordinating the work.
>
> | System Type | Vanilla | AutoTransform | AutoInject |
> |---|---|---|---|
> | Graph | 28.00 | 20.00 | 16.00 |
> | Star | 36.00 | 30.00 | 28.00 |
>
> ## **[Reference 1]** More related work in 2024-2025.
>
> We have thoroughly cited and compared relevant **papers in 2024-2025 [1-9]** in our paper, according to all reviewers’ suggestions. In short, the NetSafe is the most relevant. While they provide amazing findings about the impact of different numbers of faulty agents, they do not investigate **more subjective tasks** like translation and text evaluation. We include code generation task, enabling us to study the impact of **error types.**
>
> Additionally, our proposed AutoTransform and AutoInject methods can be viewed as **automated evaluation tools** for simulating failure scenarios and systematically measuring the robustness of multi-agent systems. This aligns with recent trends in automatic evaluation techniques (e.g., Zhang et al., 2024; Ju et al., 2024) where LLMs are used to both generate perturbations and evaluate outcomes.
>
> [1] Zhang et al., Multi-agent Architecture Search via Agentic Supernet. 2025.
>
> [2] Yu et al., A Survey on Trustworthy LLM Agents: Threats and Countermeasures. 2025.
>
> [3] Mao et al., AgentSafe: Safeguarding Large Language Model-based Multi-agent Systems via Hierarchical Data Management. 2025.
>
> [4] Zhou et al., CORBA: Contagious Recursive Blocking Attacks on Multi-Agent Systems Based on Large Language Models. 2025.
>
> [5] Wang et al., G-Safeguard: A Topology-Guided Security Lens and Treatment on LLM-based Multi-agent Systems. 2025.
>
> [6] Yu et al., LLM-Virus: Evolutionary Jailbreak Attack on Large Language Models. 2025.
>
> [7] Zhang et al., G-Designer: Architecting Multi-agent Communication Topologies via Graph Neural Networks. 2024.
>
> [8] Yu et al., NetSafe: Exploring the Topological Safety of Multi-agent Network. 2024.
>
> [9 Tran et al., Multi-Agent Collaboration Mechanisms: A Survey of LLMs. 2025.
>
> [Zhang et al. (already in our paper)] PsySafe: A Comprehensive Framework for Psychological-based Attack, Defense, and Evaluation of Multi-agent System Safety.
>
> [Ju et al. (already in our paper)] Flooding Spread of Manipulated Knowledge in LLM-Based Multi-Agent Communities.
>
> ## **[Reference 2]** Improving the introduction section.
>
> We appreciate that you like our basic ideas. We have moved the core claims to the beginning of our introduction section. The text is shown below:
>
> > Just as human organizations rely on resilient structures to function despite internal errors, multi-agent systems built on large language models (LLMs) must also withstand faulty participants to remain effective. In this paper, we argue that the resilience of LLM-based multi-agent collaboration critically depends on the system’s structural design. As these agents increasingly take on complex, collaborative tasks, their decentralized nature makes them vulnerable to clumsy or even malicious agents—those that degrade performance or disrupt workflows. Drawing a parallel to organizational theory, we find that hierarchical structures—common in robust human institutions—offer superior resilience over flat or linear ones. To rigorously test this hypothesis, we propose two methods for simulating faulty agents, and demonstrate that structural design, combined with simple defensive strategies, can significantly mitigate performance degradation.

---

### Official Review · Reviewer_iwu4 · 2025-03-13

**Overall Recommendation:** 3

**Summary:**

The paper investigates the resilience of different multi-agent architectures to faulty agents. Two approaches, autotransform, which transforms the system prompt of the agent into a malicious one, and autoinject, which takes the outputs of other agents and intentionally injects specific errors, have been proposed to create faulty agents for the evaluation. The experiments involve four different tasks and six agents. The results show that hierarchical design of the multi-agent system is more resilient to faulty agents.

**Claims And Evidence:**

"We are the first to examine how different structures of multi-agent systems affect resilience when faulty agents exist and disrupt collaboration."

There are other works a few months ahead studying the same problem, with similar results.

"The hierarchical structure has a higher resilience than the other two, exhibiting the smallest accuracy drop."

Intuitively, I tend to believe this is correct. However, beyond the structure, how the agents communicate with each other also influences the robustness. The results in Figure 3 involve two agents per type, but without the distinction of the communication scheme, which weakens the conclusion.

**Essential References Not Discussed:**

[1] Zhang et al, G-Designer: Architecting Multi-agent Communication Topologies via Graph Neural Networks, Oct 2024.
[2] Yu et al, NetSafe: Exploring the Topological Safety of Multi-agent Network, Oct 2024.

These two papers are highly relevant and were posted three months before the submission deadline. Especially in [2], there are similar conclusions regarding the relation between safety and agent topology.

**Experimental Designs Or Analyses:**

See Methods And Evaluation Criteria

**Methods And Evaluation Criteria:**

I am concerned about the design of autotransform, which assumes that the single agent functions with one piece of system prompt. In fact, the agent can interact with the environment and use tools in various ways. The system prompts can be diverse and involve multihop interaction with either the user or the environment. It is unclear whether the approach can be generalized to more general agent settings.

Also, the evaluation focuses on code generation, math problems, translation, and text evaluation. All these tasks can be achieved using an agent or just a model. Tasks not achievable by a standalone model should be considered.

**Other Comments Or Suggestions:**

n.a.

**Other Strengths And Weaknesses:**

The paper is well-written, and the motivation is well-explained. However, the scope of the work needs to be expanded. Comparison with previous works should be included.

**Questions For Authors:**

When the core LLM of one of the agent is changed to large reasoning models with a lot of (perhaps redundant) reasoning outputs, will the performance be affected?

**Relation To Broader Scientific Literature:**

Zhang et al, G-Designer: Architecting Multi-agent Communication Topologies via Graph Neural Networks, Oct 2024.

This paper studied different topologies of multi-agent systems.

Yu et al, NetSafe: Exploring the Topological Safety of Multi-agent Network, Oct 2024.

This paper studies which topology of multi-agent system is safer, which is highly related to the current work.

**Theoretical Claims:**

There are no theoretical results in this work.

---

> ### Author Rebuttal · Authors · 2025-04-01
>
> We deeply thank reviewer iwu4 for reviewing and appreciate the suggestions. We are encouraged that you find our paper well-written and sufficiently motivated! We address your concerns one by one:
>
> ## **[Claim & Evidence 1] [Broader Literature] [Reference] [W1]** Missing papers.
>
> Thanks for providing the missing references. We have thoroughly cited and compared the works of Zhang and Yu in our Related Work section. Additionally, we have included discussions about other relevant papers [1-6]. In short, the NetSafe is the most relevant. While they provide amazing findings about the impact of different numbers of faulty agents, they do not investigate **more subjective tasks** like translation and text evaluation. We include code generation task, enabling us to study the impact of **error types.**
>
> Additionally, according to ICML 2025 Reviewer Instructions (icml.cc/Conferences/2025/ReviewerInstructions), “Authors cannot expect to discuss other papers that have only been made publicly available within **four months** of the submission deadline.” Thus the two papers submitted in October are **concurrent work of our submission.**
>
> [1] Zhang et al., Multi-agent Architecture Search via Agentic Supernet.
>
> [2] Yu et al., A Survey on Trustworthy LLM Agents: Threats and Countermeasures.
>
> [3] Mao et al., AgentSafe: Safeguarding Large Language Model-based Multi-agent Systems via Hierarchical Data Management.
>
> [4] Zhou et al., CORBA: Contagious Recursive Blocking Attacks on Multi-Agent Systems Based on Large Language Models.
>
> [5] Wang et al., G-Safeguard: A Topology-Guided Security Lens and Treatment on LLM-based Multi-agent Systems.
>
> [6] Yu et al., LLM-Virus: Evolutionary Jailbreak Attack on Large Language Models.
>
> ## **[Claim & Evidence 2]** The communication scheme of agents.
>
> In our experiment, all of our selected models adopt a communication scheme of **direct messaging**, meaning that only the recipient agent will be able to see the message and no other message processing schemes such as summarizing or broadcasting is involved in the process. Therefore, the communication scheme is a **controlled variable in our experiment, and does not influence our conclusion on the system structures.**
>
> ## **[Method & Evaluation 1]** Whether AutoTransform can be generalized to more general agent settings (tool uses).
>
> AutoTransform takes the whole agent profiles as the subject of our transformation. While system prompts can be diverse (for multiple tool uses), **an agent’s profile is usually consistent**, since it includes the main goals, task descriptions and constraints of an agent. Therefore, our method can work with agents equipped with tools that can interact with the environment.
>
> Moreover, some of our selected systems **are already equipped with external tools**. For instance, Self-collab equipped agents with a python interpreter for code testing, and MetaGPT’s agents have access to a message pool from which to retrieve history messages when performing tasks. This ensures our method is still valid in the more general settings. We also compute the performance on code tasks of different systems by whether or not they have tool modules. As is shown below, **the performance of AutoTransform is consistent with these two types of agent systems.**
>
> | System Type | Vanilla | AutoTransform |
> |---|---|---|
> | w/o Tool Use | 63.20 | 47.00 |
> | w/ Tool Use | 66.27 | 42.70 |
>
> ## **[Method & Evaluation 2]** Other tasks.
>
> We acknowledge the need for evaluating more complex, real-world tasks hard to complete by a standalone model, such as making complete, executable software. For example, generate **a snake game with pygame**. Using a single GPT-3.5, **the code is incomplete and the snake cannot be controlled**. Using CAMEL, we achieve an **executable game**, with extensive features.
>
> When using **AutoTransform**, the code is **still executable** but the snake's **speed is too fast and the mapping between arrow key and snake's movement is mixed** (e.g., pressing "left" will make the snake move upward). When using **AutoInject**, the game is totally **unexecutable** due to multiple errors injected. Our method is still effective in such real-world scenarios.
>
> Additionally, though the tasks can be completed by single agents, from the results of single agents vs. multi-agents in Figure 4 on page 6 we can see that **multi-agent systems can improve the performance over single agents.**
>
> ## **[Q1]** CoT models.
>
> We explore the impact of CoT using a reasoning model, **o1-mini, on the Math task** due to budget limits. The results are presented in the tables below. For models using CoT, the performance is largely improved. However, we can still observe the **performance decrease** caused by faulty agents. **Hierarchical structures still outperform the other two.**
>
> | o1-mini (Math) | Linear | Flat | Hierarchical |
> |---|---|---|---|
> | No Attack | 78.00 | 80.00 | 81.00 |
> | AutoTransform | 14.00 | 18.00 | 63.00 |
> | AutoInject | 70.00 | 74.00 | 77.00 |

---

### Official Review · Reviewer_xvqz · 2025-03-14

**Overall Recommendation:** 3

**Summary:**

This paper investigates the resilience of large language model (LLM)-based multi-agent systems against faulty or malicious agents. It compares different system architectures—Linear, Flat, and Hierarchical—across tasks like code generation, math problem-solving, translation, and text evaluation, finding hierarchical structures most resilient. Additionally, it introduces methods to simulate agent errors (AUTOTRANSFORM, AUTOINJECT) and proposes defense strategies (Challenger, Inspector) that significantly enhance system robustness.

**Claims And Evidence:**

Overall, the paper's claims are well-supported by clear experimental results and analyses. Key findings—such as hierarchical structures offering better resilience, semantic errors causing greater impact than syntactic ones, and certain error injections even improving performance—are convincingly demonstrated. However, the explanation for why AUTOTRANSFORM errors have less impact than AUTOINJECT in GPT-3.5 contexts may need further empirical validation or deeper analysis.

**Essential References Not Discussed:**

N/A

**Ethical Review Concerns:**

Since this research is about multiagent attack and defense, ethical review is required

**Ethical Review Flag:**

Flag this paper for an ethics review.

**Ethics Expertise Needed:**

["Privacy and Security"]

**Experimental Designs Or Analyses:**

In your experiment, you using different multiagent framework for those structure, metagpt, camel etc, how do you make sure it is gonna be a fair comparison of those structures then, since they all use different system prompt and some special design, so I would question the fairness when you using 6 different code base for the test to represent for different structures.

**Methods And Evaluation Criteria:**

In general sense the methods make sense.

When you specify the reasons at the line 261-268, you should specify proofs of those such as citations, or giving examples, similarly from line 247 to 253, you should give more proof instead of just giving a reason with out proof.

**Other Comments Or Suggestions:**

**Suggestion**:

1.  I would like to see some other communication topologies can be considered such as Tree, star, graph, in Scaling Large Language Model-based Multi-Agent Collaboration

2. I would like to see how your method will affect  single agent, then you can. Compare with how multiagent can handle it better or worse

**Other Strengths And Weaknesses:**

**Strengths**:

1. The paper addresses an important and timely research topic.

2. It represents the first systematic exploration within this specific area.

**Weaknesses**:

1. The graphical presentation could be improved for better readability and clarity. Some explanations require refinement; for example, on line 19, clearly define what agents A, B, and C represent, ideally adopting a more formal mathematical notation.

2. The explanations provided between lines 261–268 and 247–253 lack sufficient empirical support or citations. The authors should strengthen their claims by providing concrete examples or referencing relevant prior studies.

3. When presenting results and findings, explicitly include numerical data and specific evidence to substantiate conclusions clearly.

**Questions For Authors:**

N/A

**Relation To Broader Scientific Literature:**

The paper related to prompt attack agent attack, agent system roboustness

**Theoretical Claims:**

N/A

---

> ### Author Rebuttal · Authors · 2025-04-01
>
> We deeply thank reviewer xvqz for your time and effort in reviewing our work, and your invaluable comments that further enrich our paper. We are particularly encouraged that you find our claims well-supported, our methods & evaluations reasonable, and by your recognition of the importance and novelty of our work. We address your concerns here:
>
> ## **[Claim & Evidence]** AutoTransform is harder to control than AutoInject.
>
> The reason why faulty agents produced by AutoTransform have less impact than directly injecting errors (AutoInject) is because GPT-3.5-Turbo is **weaker in terms of the precise control of generated errors**. We conduct an analysis using AutoTransform to instruct agents to introduce syntax errors in 20% and 40% of the code lines. The results are summarized below:
>
> | Error Rate | Avg | Std | Min | Max |
> |---|---|---|---|---|
> | Instruct 20% | 1.56 | 3.65 | 0.00 | 14.30 |
> | Instruct 40% | 9.49 | 26.70 | 0.00 | 90.10 |
>
> These results indicate **significant variability**, with agents struggling to consistently achieve the precise error rates of 20% or 40%. This underscores the necessity and robustness of our AutoInject method.
>
> ## **[Method & Evaluation] [W2 & W3]** Inclusion numerical data and specific evidence to support our claims.
>
> We have added numbers at (1) Left-column: Line 248, 251, 261, 274 and (2) Right-column: Line 251, 253, 257, 263, 292, 308, from the results in Table 5, 6, 7, and 8 in Appendix C on page 13.
>
> ## **[Design & Analysis]** Fair comparison using the 6 different code bases.
>
> Different structures have various role designs. For example, **a flat system may not have a leader, where the other two systems have**. There is not a set of profiles applicable to all multi-agent systems. Therefore, instead of manually writing profiles, we use the code bases from published studies which have been optimized by the authors for the specific structures. To offer fairer comparison, i.e., to mitigate the impact of using different prompts, we have included two systems per structure.
>
> ## **[W1]** Math definition on agent collaborations.
>
> A multi-agent system can be defined as a graph: $G = (V, E)$, where $V$ represents agents and $E \subseteq V \times V$ is a set of directed edges. Each $(u, v) \in E$ denotes agent $u$ reports to agent $v$.
>
> - Linear systems are: **directed path graphs**, where $\forall v \in V, v \neq s, v \neq t$, we have: $\deg^+(v) = \deg^-(v) = 1$; for the endpoints, $\deg^-(s) = 0$, $\deg^+(s) = 1$, $\deg^+(t) = 0$, and $\deg^-(t) = 1$. Agents in this structure form a chain from $s$ to $t$.
>
> - Flat systems are: **directed complete graphs** with bidirectional edges, where $\forall u, v \in V, u \neq v$, both $(u, v) \in E$ and $(v, u) \in E$. This represents a fully connected, non-hierarchical structure.
>
> - Hierarchical systems are: **rooted directed trees**, where there exists a unique root agent $r \in V$ such that $\deg^-(r) = 0$, and $\forall v \in V \setminus {r}$, $\deg^-(v) = 1$. The structure is acyclic and forms a strict top-down hierarchy.
>
> ## **[Q1]** Other communication topologies.
>
> Inspired by MacNet and GPTSwarm, we design two advanced graph-based multi-agent frameworks using four agents:
>
> 1. **Complete graph (a flat structure)**: each agent generates their own answers. After receiving all others’ answers in the next run, they re-generate the answers after thinking. The final answer is the majority one.
>
> 2. **Star (a hierarchical structure)**: one leader proposes three approaches and distributes them to the three agents. After receiving the solutions, the leader gives its evaluation and generates the final answer.
>
> We evaluate GPT-3.5 using the Math task. The performance is shown in the table below. We can conclude that our methods and analyses are **applicable to diverse frameworks**. Flat structure still has **a lower performance** since there is no leader coordinating the work.
>
> | System Type | Vanilla | AutoTransform | AutoInject |
> |---|---|---|---|
> | Graph | 28 | 20 | 16 |
> | Star | 36 | 30 | 28 |
>
> ## **[Q2]** Impact on single agents.
>
> We conduct experiments on applying the two error-introducing methods on **a single agent based on GPT-3.5-Turbo** across all four tasks. The performance is shown below:
>
> | Tasks | Vanilla | AutoInject | AutoTransform |
> |---|---|---|---|
> | Code | 58.41 | 15.24 | 3.92 |
> | Math | 24.00 | 18.00 | 8.00 |
> | Translate | 68.42 | 61.08 | 68.42 |
> | Text Eval | 41.25 | 32.50 | 18.75 |
>
> Compared to the performance of other multi-agent systems (the table below), we conclude that **all three types of systems have better resilience** against both methods compared to a single agent. This is because the systems have other “good” agents for reviewing and testing, which can identify the errors made by the faulty agent.
>
> | Systems | Vanilla | AutoInject | AutoTransform |
> |---|---|---|---|
> | Single Agent | 48.02 | 31.71 | 21.66 |
> | Linear | 55.62 | 38.27 | 38.24 |
> | Flat | 54.37 | 40.25 | 43.93 |
> | Hierarchical | 53.00 | 48.12 | 46.57 |

---

### Official Review · Reviewer_LJmt · 2025-03-25

**Overall Recommendation:** 3

**Summary:**

This paper explores the resilience of multi-agent collaboration by introducing faulty agents and errors. It embarks on an empirical approach to examine performance drops in different scenarios / multi-agent system structures. The authors introduce AUTOTRANSFORM and AUTOINJECT algorithms for creating faulty agents with the evaluation based on error types, quantities, and frequencies. The work provides some insights on the impact of faulty agents across different tasks, including code generation, translation, math and text evaluation.

**Claims And Evidence:**

The paper provides comprehensive empirical experiments with multiple tasks (code generation, math solving, translation and text evaluation), system structures (linear, flat and hierarchical) and error types (semantic and synaptic). The impacts of faulty agents are observed in various settings and illustrated in different figures.

**Essential References Not Discussed:**

Some missing references:
- Perez, E., Huang, S., Song, F., Cai, T., Ring, R., Aslanides, J., ... & Irving, G. (2022). Red teaming language models with language models. arXiv preprint arXiv:2202.03286. - vulnerabilities in LLMs.
- Tan, S., Joty, S., Baxter, K., Taeihagh, A., Bennett, G. A., & Kan, M. Y. (2021). Reliability testing for natural language processing systems. arXiv preprint arXiv:2105.02590. - reliability testing for NLP systems
- Tran, K. T., Dao, D., Nguyen, M. D., Pham, Q. V., O'Sullivan, B., & Nguyen, H. D. (2025). Multi-Agent Collaboration Mechanisms: A Survey of LLMs. arXiv preprint arXiv:2501.06322. - multi-agent collaboration

**Experimental Designs Or Analyses:**

The experimental design of the paper is reasonably well-executed, with multiple experiments across different multi-agent systems and diverse tasks. The authors also provide various ways of controlling errors to investigate the performance impacts.

**Methods And Evaluation Criteria:**

The proposed methods are reasonable for investigating the resilience of multi-agent AI system with two algorithms: AUTOTRANSFORM and AUTOINJECT. It also provide a number of tasks, including coding, math solving, translation and text evaluation for better generalizability of its findings.

**Other Comments Or Suggestions:**

Nil

**Other Strengths And Weaknesses:**

Strengths:
- Comprehensive evaluations across multiple dimensions: tasks, structures, error types and rates, providing the understanding and benchmarks of resilience in mult-agent AI systems.
- Clear implications for designing robust MAS with findings on various multi-agent structures.
- Clear explanations of observations and findings, relating the performance impacts to collaboration mechanisms.

Weaknesses:
- Lack of theoretical analyses/formal proofs and deeper investigations on collaboration mechanisms.
- Faulty agents in the world might be more complicated with dynamic scenarios among multiple agents.
- Only GPT models were investigated without the use of chain-of-thought abilities (which might be useful in improving resilience)
- The insights of multi-agent structures are limited as there should be mechanisms to improve existing structures.

**Questions For Authors:**

- Can the authors explore dynamic scenarios with multiple faulty agents? There should be a way to control the number of agents and the functionality-to-errors ratio in each agent to be more realistic.
- Can you explain why certain structures are more resilient, thereby leading to better configurations or optimization of the agent systems?
- Noises can also lead to better performance, can the authors explore further the key mechanisms behind these observations?
- What are the practical implications and broad impacts of your paper?

**Relation To Broader Scientific Literature:**

- The paper broadly relates to LLM-based multi-agent collaboration, building on existing literature work, such as MetaGPT, AgentVerse, etc. The authors investigate the concept of resilience in these multi-agent systems by introducing faulty agents.
- It also provides a theoretical basis for understanding multi-agent collaboration topology, demonstrating certain structures perform better.
- The key contributions of the paper extend the broad areas of AI safety and trustworthiness.

**Theoretical Claims:**

The paper does not have mathematical proofs.

---

> ### Author Rebuttal · Authors · 2025-04-01
>
> We deeply appreciate reviewer LJmt’s time for reviewing and providing valuable suggestions. We are encouraged that you find our experiments comprehensive and well-executed, methods reasonable, and conclusions helpful for the community. We address your concerns here:
>
> ## **[W1]** Math definition on agent collaborations.
>
> A multi-agent system: $G = (V, E)$, where $V$: agents and $E \subseteq V \times V$: each $(u, v) \in E$ denotes agent $u$ reports to agent $v$.
>
> - Linear systems are: **directed path graphs**, where $\forall v \in V, v \neq s, v \neq t$, we have: $\deg^+(v) = \deg^-(v) = 1$; for the endpoints, $\deg^-(s) = 0$, $\deg^+(s) = 1$, $\deg^+(t) = 0$, and $\deg^-(t) = 1$.
>
> - Flat systems are: **directed complete graphs**, where $\forall u, v \in V, u \neq v$, both $(u, v) \in E$ and $(v, u) \in E$.
>
> - Hierarchical systems are: **rooted directed trees**, where there exists a unique root agent $r \in V$ such that $\deg^-(r) = 0$, and $\forall v \in V \setminus {r}$, $\deg^-(v) = 1$.
>
> ## **[W2]** Real-world complicated scenario.
>
> To generate **a snake game with pygame**. Using a single GPT-3.5, **the code is incomplete and the snake cannot be controlled**. Using CAMEL, we achieve an **executable game**.
>
> When using **AutoTransform**, the code is still executable but the mapping between arrow key and snake's movement is mixed. When using **AutoInject**, the game is totally unexecutable. Our method is still effective in such real-world scenarios.
>
> ## **[W3]** Other models & CoT.
>
> We conduct experiments using the LLaMA-3.1-70B-Instruct model on all the four tasks and a reasoning model, o1-mini, on the Math task. The results are presented in the tables below. The hierarchical structures perform the best still hold for non-GPT-based models. The performance is largely improved with CoT (o1), while we can still observe the impact of faulty agents. Hierarchical structures still outperform the other two.
>
> | LLaMA | Linear | Flat | Hierarchical |
> |---|---|---|---|
> | No Attack | 73.78 | 76.83 | 76.15 |
> | AT | 11.90 | 39.03 | 66.96 |
> | AI | 38.72 | 36.59 | 55.64 |
>
> | o1 | Linear | Flat | Hierarchical |
> |---|---|---|---|
> | No Attack | 78 | 80 | 81 |
> | AT | 14 | 18 | 63 |
> | AI | 70 | 74 | 77 |
>
> ## **[W4]** Advanced structures.
>
> Inspired by MacNet and GPTSwarm, we design two advanced graph-based multi-agent frameworks using four agents:
>
> 1. **Complete graph (a flat structure)**: each agent generates their own answers. In the next run, they re-generate the answers after thinking on others' results.
>
> 2. **Star (a hierarchical structure)**: one leader proposes three approaches and distributes them to the three agents and generates the final answer.
>
> We evaluate GPT-3.5 using the Math task. Our methods and analyses are **applicable to diverse frameworks**. Flat structure still has **a lower performance**.
>
> | System Type | Vanilla | AT | AI |
> |---|---|---|---|
> | Graph | 28 | 20 | 16 |
> | Star | 36 | 30 | 28 |
>
> ## **[Q1]** Multiple faulty agents.
>
> We explore the scenario of two faulty agents in AgentVerse on the Math task using GPT-3.5. AgentVerse has 4 agents, with Solver faulty by default. Here, we make an additional one, either Critic or Planner, faulty. Our methods are still valid. The Planner, who decides the high-level direction, plays a more important role. Its faults cause a greater performance decrease.
>
> | Faulty Agents | Error Method | Performance |
> |---|---|---|
> | None | None | 28 |
> | Solver | AI | 20 |
> | S + Critic | AI | 14 |
> | S + Planner | AI | 12 |
> | S | AT | 16 |
> | S + Critic | AT | 14 |
> | S + Planner | AT | 2 |
>
> ## **[Q2]** Explanation on performance of different structures.
>
> We can have an analogy with real-world human organization structure. A hierarchical structure enables centralized decision-making, where a top-level role gathers information and efficiently distributes decisions through clear chains of command. In contrast, a flat structure often lacks clear leadership, leading to decision paralysis and coordination issues. A linear structure has a defined chain of command, but communication is slower, and top leaders have limited oversight of lower levels.
>
> ## **[Q3]** Explanation on how noises improve performance.
>
> (1) Double Checking: Injecting obvious errors prompts agents to respond with corrections, which often fix both the injected and existing issues. (2) Divergent Thinking: Systems can stagnate due to repetitive reasoning from identical LLMs. Introducing major errors shifts the discussion, promoting fresh insights.
>
> ## **[Q4]** Practical implications.
>
> (1) designing hierarchical multi-agent systems help, which reflects a prevalent collaboration mode in real-world human society. (2) Agents can have questions or suggestions on others' results, boosting fault recovery.
>
> ## **[References]** Missing papers.
>
> We have cited and compared the works of Perez, Tan, and Tran in our paper. In short, we consider a new scenario where the vulnerability comes from weaker agents in a multi-agent system.

---

> > ### Comment · Reviewer_LJmt · 2025-04-04
> >
> > Thanks for addressing my comments. I have updated my score.

---

> > > ### Author Response · Authors · 2025-04-04
> > >
> > > We deeply thank reviewer LJmt for checking our response! We are glad that your concerns are addressed. Your comments are important for further improving this work. Thanks once again for re-considering your rating!

---

### Decision · Program_Chairs · 2025-05-01

**Decision:**

Accept (poster)

**Comment:**

This paper presents a systematic and timely investigation into the resilience of large language model (LLM)-based multi-agent systems when subject to faulty agents. Through the design and application of two novel error injection methods—AUTOTRANSFORM and AUTOINJECT—the authors simulate semantic and syntactic disruptions in agents' behavior across various collaborative settings. The experiments, conducted over diverse tasks (code generation, math solving, translation, and text evaluation) and multi-agent topologies (linear, flat, and hierarchical), consistently demonstrate that hierarchical structures are the most robust to such failures (Reviewers 1, 2, 3, 4). The work contributes empirical insights into the safety and robustness of collaborative AI, supported by well-executed experimental protocols and meaningful metrics. Reviewers also praise the paper's relevance to the broader literature on LLM agents and its practical implications for designing trustworthy AI systems (1, 2, 4).

Weaknesses and Limitations:
Despite its strengths, reviewers highlighted several limitations that should be addressed in future revisions. The lack of theoretical analysis and formal proofs (1, 3) reduces the generalizability of some claims, and the evaluation is constrained by a reliance on only GPT-based models, excluding more diverse LLMs or reasoning strategies like chain-of-thought (1). Several reviewers questioned the fairness of comparing architectures based on different frameworks and system prompts (2, 3), noting that communication protocols and prompting strategies could confound results. Furthermore, important recent works on agent topology and safety were omitted from the citations (1, 3, 4), and the discussion could benefit from integration with other architectures like LangGraph or GPTSwarm (4). Lastly, reviewers suggest that the scope of agent fault modeling could be expanded to consider more dynamic or complex failure scenarios (1, 3). Nonetheless, these issues are outweighed by the novelty, execution quality, and potential impact of the work, supporting a positive acceptance recommendation.